# XD: Cross-lingual Knowledge Distillation for Polyglot Sentence Embeddings

## Abstract

Current state-of-the-art results in multilingual natural language inference (NLI) are based on tuning XLM (a pre-trained polyglot language model) separately for each language involved, resulting in multiple models. We reach significantly higher NLI results with a single model for all languages via multilingual tuning. Furthermore, we introduce cross-lingual knowledge distillation (XD), where the same polyglot model is used both as teacher and student across languages to improve its sentence representations without using the end-task labels. When used alone, XD beats multilingual tuning for some languages and the combination of them both results in a new state-of-the-art of 79.2% on the XNLI dataset, surpassing the previous result by absolute 2.5%. The models and code for reproducing our experiments will be made publicly available after de-anonymization.

## 1 Introduction

Many sentence level tasks in natural language processing have seen efficient solutions based on sentence vector representations (embeddings) and supervised tuning, however labelled data is scarce for all but a handful of resource-rich languages like English. This motivates the development of cross-lingual methods that can perform knowledge transfer from sentence representations in one language to labels assigned to sentences in another language.

We focus on one such task: natural language inference (NLI), where the aim is to detect, whether the meaning of one sentence can be inferred from another one, contradicts it, or neither. For instance, the sentence *You can leave* can be inferred from the sentence *You don't have to stay there*. The example is taken from the XNLI dataset (Conneau et al., 2018), which was created for testing cross-lingual NLI and includes labelled English sentence pairs, translated into 15 languages; manually for the development and test sets, automatically for the training set.

The best results on XNLI so far have been achieved by using XLM (Lample & Conneau, 2019), a contextualized multilingual language model that is pre-trained on unlabelled text and then tuned in a supervised manner separately for each language in the XNLI dataset. While this yields competitive results, it wastefully employs a separate NLI model for each language (Lample & Conneau, 2019).

Our contributions are two-fold: first we describe a multilingual tuning scenario, in which we achieve a significantly higher average XNLI accuracy with a single model for all 15 languages. Furthermore, we introduce XD, a cross-lingual knowledge distillation approach that uses one and the same XLM model to serve both as teacher (for English sentences) and student (for their translations into other languages). The approach does not require end-task labels and can be applied in an unsupervised setting. We describe the relevant background and our methods in Section 2.

In the experimental part of this paper (Section 3) we compare the performance of XD and multilingual tuning for multiple language combinations, in order to cover low and high resource settings as well as related and distant language pairs. Our results show that XD reaches the same or better results than multilingual tuning alone, depending on language. A combination of both methods brings an even better XNLI average result, which outperforms both of them alone and surpasses the previous state-of-the-art as well as results of concurrent work.

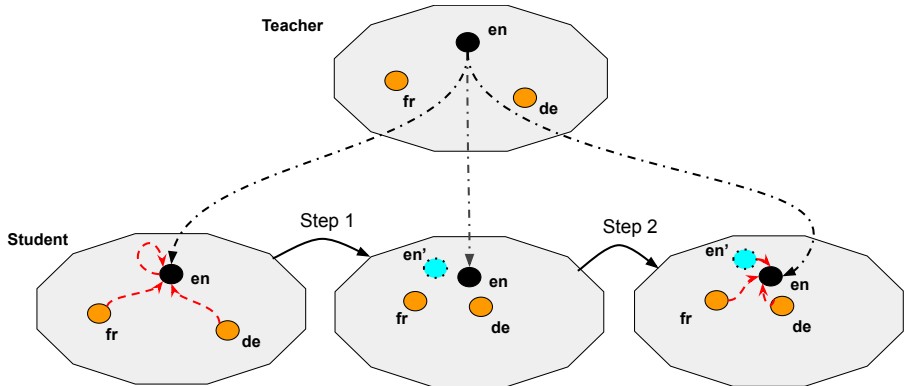

Figure 1: An illustration of cross-lingual knowledge distillation (XD). Before the first step the same polyglot model is used as teacher and student. At the first step the student model adapts its representation of other languages to English via translation examples (red arrows). Then the representations for other languages become closer to English in the latent logits space (orange dots). However, since the same model operates on English as well, representations for English also change and can not serve as optimal targets anymore (purple dot). That is why we employ teacher network to produce gold target for the next step (black dot). At the last step we continue alignment to the original English target provided by the teacher model. We repeat XD until convergence.

## 2 METHODOLOGY

### 2.1 BACKGROUND

We build both our methods (described in the following subsections) based on a large-scale pre-trained cross-lingual language model (XLM) introduced by Lample & Conneau (2019). XLM is trained either with the Masked Language Modeling (MLM) objective alone or in combination with the Translation Language Modeling (TLM) objective. In MLM (Devlin et al., 2018) we mask words in the input sequence and teach the model to fill in the gaps. In $\text{XLM}_{MLM}$ we train a model on sentences from different languages by employing joint multilingual wordpiece vocabulary (Sennrich et al., 2015). Shared cross-lingual wordpieces lead to the sharing of cross-lingual wordpiece embeddings. In $\text{XLM}_{MLM+TLM}$ setting we train the model on pairs of parallel sentences while masking words similarly to $\text{XLM}_{MLM}$. However in $\text{XLM}_{MLM+TLM}$ model can attend not only to surrounding context to predict the missing word, but also to its translation from parallel sentence which results in stronger cross-lingual signal. Both methods belong to the pretraining time cross-lingual alignment category.

After the language model is trained, we substitute the word prediction head with classification layer (3-class softmax in our case). We place this head on top of the contextual embedding of special "CLS" token prepended to the sentence. Next, we can tune result model on source language training data and directly apply it to target languages obtaining zero-shot classification results. By the source language we mean that language we have labeled training data and by target languages we mean the languages that we try to transfer knowledge to (languages without out-of-the-box supervised data). On the other hand, absolute state-of-the-art result is obtained by translating English portion of the data into target languages with MT system(s), duplicating pretrained $\text{XLM}_{MLM+TLM}$ as many times as there are target languages, and tuning on each language individually (Lample & Conneau, 2019).

Another concept important for understanding our method is knowledge distillation (Hinton et al., 2015). In the knowledge distillation framework we train so-called student model to mimic behaviour of the (usually larger and stronger) teacher model. In case of classification, one way to perform KD is to make teacher model predict on unlabeled examples producing vector of real-valued logits (unnormalized probabilities over discrete label space). Student model then takes the same example as its input and uses teacher's logits as its target.

Table 1: BLEU scores of the machine translation system used for generating synthetic training data for multilingual tuning and knowledge distillation corpus for XD. The table is taken from (Conneau et al., 2018)

| $l_2$ | fr | es | de | el | bg | ru | tr | ar | vi | th | zh | hi | sw | ur |
|---|---|---|---|---|---|---|---|---|---|---|---|---|---|---|
| En-$l_2$BLEU | 49.3 | 48.5 | 38.8 | 42.4 | 34.2 | 24.9 | 21.9 | 15.8 | 39.9 | 21.4 | 23.2 | 37.5 | 24.6 | 24.1 |

## 2.2 MULTILINGUAL TUNING

In multi-language fine-tuning, similarly to current state-of-the-art approach, we use machine translation system to translate data to other languages. Then we however use all the obtained data at the same time to tune a single $XLM_{MLM}$ or $XLM_{MLM+TLM}$. Since the pseudo corpus is fully aligned, model does not see a single new example. However, the network might improve by discovering task-oriented rules and regularities learnt between languages.

## 2.3 XD: CROSS-LINGUAL KNOWLEDGE DISTILLATION

We propose to employ knowledge distillation method to perform cross-lingual transfer in a single cross-lingual language model. We duplicate the network twice and mark original one as a student and its copy as teacher model. Then for each target language we use parallel source-target text pairs as following:

1. we pass source text to the teacher model to get continous logits vector representation;
2. we pass target text to the student model to get continous logits vector representation;
3. we compute L2 loss between teacher logits and student logits
4. we backpropagate through student while keeping teacher frozen

Above-mentioned knowledge distillation procedure is based on the idea that the sentence should get same latent space representation no matter in what language it was proxied to the model (Conneau et al., 2018; Aldarmaki & Diab, 2019a). Since our polyglot language model was tuned on labeled downstream task data for English language we align all other language representations to the English latent space.

Using the same *polyglot* model as teacher and student brings us following benefits:

- we do not need to train a separate student model for each target language; instead we just align cross-lingual representations from the same model;
- we do not need to train a separate teacher model for source language; instead we just operate on the source language logits latent space of our polyglot language model;
- we get a decent student initialization where representations for source and target text are already close to each other to some degree which simplifies and speeds up convergence

Also knowledge distillation procedure abolishes the need of using target language labels for cross-lingual alignment.

Multilingual tuning can also be viewed from the knowledge distillation perspective. Teacher logits for a given sentence can be viewed as continuous (approximate) representation for this sentence. True sentence's label can be viewed as discrete representation for this sentence. One important conceptual difference is that in the former case, different sentences belonging to the same class will get the same discrete representation (same label). In the latter case, different sentences belonging to the same class will get different continuous space (logits) representations. Continuous targets provide richer but at the same time more restrictive signal. Depending on the strength of teacher model this might lead or not lead to the better training comparing to learning from discrete vectors. The quality of parallel corpora as well as languages relatedness are another important factors that emerge in the context of XD.

From the transfer learning perspective XD method belongs to the sequential learning paradigm (Ruder, 2019) because we first pretrain language model, then fine-tune it repeatedly. It also can

Table 2: Results of multilingual tuning (MLT) for all languages, as well as subsets of languages: removing German/Swahili/Urdu, compared with individual tuning (IndT, results of tuning a separate model for each language by Lample & Conneau (2019)). Results for the two XLM varieties (MLM and TLM) are shown separately. Zero-shot scores (with no directly supervised tuning performed for these languages) are shown in gray.

| | en | fr | es | de | el | bg | ru | tr | ar | vi | th | zh | hi | sw | ur | avg |
|---|---|---|---|---|---|---|---|---|---|---|---|---|---|---|---|---|
| $\text{XLM}_{MLM}$ | | | | | | | | | | | | | | | | |
| Baseline | 83.7 | 76.0 | 76.8 | 73.8 | 73.2 | 73.8 | 72.1 | 66.9 | 68.6 | 71.9 | 67.6 | 71.5 | 64.3 | 64.4 | 61.3 | 70.7 |
| MLT | 83.3 | 79.4 | 80.0 | 78.5 | 78.8 | 79.2 | 76.8 | 74.2 | 75.0 | 75.8 | 75.0 | 78.0 | 71.8 | 71.5 | 66.6 | 76.6 |
| $\text{MLT}_{no-de}$ | 83.1 | 79.9 | 80.5 | 76.1 | 78.7 | 79.1 | 76.7 | 74.1 | 74.6 | 75.3 | 74.4 | 75.5 | 70.2 | 70.7 | 67.2 | 76.1 |
| $\text{MLT}_{no-sw}$ | 83.0 | 79.2 | 80.1 | 78.6 | 78.4 | 79.0 | 76.4 | 74.4 | 75.1 | 75.3 | 75.0 | 78.2 | 71.5 | 67.8 | 64.6 | 75.9 |
| $\text{MLT}_{no-ur}$ | 83.5 | 79.7 | 80.3 | 78.9 | 78.5 | 79.2 | 76.4 | 74.8 | 75.3 | 76.3 | 75.8 | 77.4 | 71.6 | 71.4 | 66.5 | 76.5 |
| $\text{XLM}_{MLM+TLM}$ | | | | | | | | | | | | | | | | |
| Baseline | 84.9 | 78.8 | 79.2 | 76.6 | 76.4 | 76.7 | 75.0 | 72.4 | 72.4 | 74.3 | 71.4 | 74.3 | 68.5 | 68.6 | 64.4 | 73.8 |
| IndT | 85.0 | 80.2 | 80.8 | 80.3 | 78.1 | 79.3 | 78.1 | 74.7 | 76.5 | 76.6 | 75.5 | 78.6 | 72.3 | 70.9 | 63.2 | 76.7 |
| MLT | 84.6 | 80.6 | 81.6 | 80.2 | 79.9 | 80.4 | 78.4 | 76.1 | 77.6 | 78.1 | 77.2 | 79.4 | 73.3 | 73.1 | 68.5 | 78.0 |
| $\text{MLT}_{no-de}$ | 85.2 | 80.9 | 81.9 | 79.5 | 79.6 | 81.0 | 79.6 | 76.5 | 77.7 | 77.2 | 77.1 | 78.3 | 74.3 | 72.7 | 70.0 | 78.2 |
| $\text{MLT}_{no-sw}$ | 84.9 | 80.3 | 81.3 | 79.9 | 79.5 | 80.2 | 77.9 | 75.5 | 77.2 | 76.9 | 76.2 | 78.9 | 72.6 | 70.4 | 66.9 | 77.2 |
| $\text{MLT}_{no-ur}$ | 85.2 | 81.3 | 82.5 | 80.5 | 80.8 | 81.2 | 79.3 | 76.7 | 77.8 | 78.5 | 77.5 | 80.2 | 73.9 | 73.0 | 70.1 | 78.9 |

be classified as cross-lingual learning since the last fine-tuning step is done for multiple languages in terms of the same polyglot model.

Next we describe the experiments in which we validate both our introduced approaches.

# 3 EXPERIMENTS AND RESULTS

## 3.1 EXPERIMENTAL SETTINGS

We choose the XNLI dataset (Conneau et al., 2018) as our test bed for evaluating models from all following experiments. XNLI provides human translated validation and test data for 15 languages that belong to different language families. Baselines are tuned on the English MNLI corpus (Williams et al., 2018) that contains more than 400k training examples. For multilingual tuning we use machine translated versions of MNLI from (Conneau et al., 2018) for a fair comparison with current state-of-the-art tuning approach (Lample & Conneau, 2019). For XD unlabeled parallel data is needed, so we use the same parallel sentences but with labels removed. The implications of this choice are two-fold. On one hand using the translated training set for XD is ideal for the teacher network to show its knowledge during distillation. On the other hand, it allows for a fair comparison with multilingual tuning because we do not show the model any new examples (but only translations of what it had already seen). Also, the parallel data are not required to be synthetic for XD and better results can likely be achieved with actual parallel data.

For all the XD experiments we use English as our source language (since there is ground-truth training data available for it) and for case studies we choose French, Swahili and Urdu as our target languages. We base our choice on quality of synthetic parallel corpus and relatedness of languages to English. The quality of translations defines how far our setting falls from what could be achieved with real parallel data and implicitly represents the resource richness of the target language. Relatedness of the languages defines the amount of shared wordpieces (Sennrich et al., 2015) which also impacts XD. French (fr) is similar to English while also being a high-resource language with a high translation BLEU score (49.3); Urdu (ur) is an unrelated language with a low BLEU score of 24.1; Swahili (sw) is loosely between French and Urdu in terms of relatedness to English but with a low BLEU score of 24.6. For multilingual tuning zero-shot results we choose German, Swahili, and Urdu for the same reasons. Refer to the Table 1 for other languages' BLEU scores.

We choose $\text{XLM}_{MLM/MLM+TLM}$ as our polyglot language model because it gave state-of-the-art cross-lingual results at the moment of writing[1]. For all XD and multilingual tuning experiments we use hyperparameters optimized for our baselines. We use the Adam optimizer with a learning

---

[1]Concurrently with our work Huang et al. (2019) presented an even stronger polyglot language model

rate of $5 \times 10^{-6}$, batch size of 8 sentences from the same language (truncating sentences to be up to 256 words per example), and small epochs of $20,000$ examples each (following Lample & Conneau (2019)). We use L2 as our loss function for XD. Together with the codebase we also publish training configuration files for all the experiments we ran where the setup can be seen in exact details. [2]. We use pretrained XLM from *pytroch-transformers*[3] repository and *allennlp*[4] as our NLP framework.

### 3.2 MULTILINGUAL TUNING

#### 3.2.1 ALL LANGUAGES

The current state-of-the art approach requires translating the training set into the target language and tuning a polyglot $XLM_{MLM+TLM}$ model on this language for each language individually (IndT). The disadvantage of this approach is that in case of N languages it requires N models to be tuned and maintained. By doing multilingual tuning (MLT) for $XLM_{MLM+TLM}$ on all the languages at the same time we not only get rid of the multiple models requirement but also push the current state-of-the-art result by 1.3 points on average. We explain this result with the fact that for many languages synthetic parallel data is of low quality and thus transfer from other languages is crucial.

We also improve by a large margin of 5.9 and 4.2 points over $XLM_{MLM}$ and $XLM_{MLM+TLM}$ models tuned only on English data. Higher increase for $XLM_{MLM}$ might be due to the fact that $XLM_{MLM+TLM}$ already has higher cross-lingual power and thus tuning on more multilingual examples might be less effective.

See Baseline (English-tuned XLM model), IndT (individually tuned XLM models), and MLT (multilingually tuned XLM model) entries in Table 2 for the full results.

#### 3.2.2 ZERO-SHOT CROSS-LINGUAL TRANSFER

In this section we inspect the power of multilingual tuning (MLT) for obtaining zero-shot results for specific languages. We do so by removing the languages from the MLT tuning scheme and evaluating zero-shot performance on these languages. As can be seen from MLT w/o in Table 2 zero-shot performance is lower for German and Swahili (comparing to all languages MLT), but higher for Urdu. This might suggest it is better to use low-BLEU data from unrelated language at all. The model learns to transfer knowledge from other languages quite effectively in both $XLM_{MLM}$ and $XLM_{MLM+TLM}$ setting. It is interesting to see that by not using German data for training the model also produces good scores for Urdu. This results suggests that wee need more linguistically driven experiments to make confident conclusions. The fact that by using MLT zero-shot we improve over English-only shows that MLT is effective for cross-lingual transfer. Note that by using translated training set data we do not introduce any new examples to the model and thus improvements come solely from cross-lingual transfer.

### 3.3 XD: CROSS-LINGUAL KNOWLEDGE DISTILLATION

#### 3.3.1 ALL LANGUAGES

In this subsection we perform XD on all languages from XNLI dataset. We also include English so that the model does not forget its original English performance. We present results in Table 3.

As can be seen XD beats XLM zero-shot baseline for in both MLM and MLM + TLM cases by 3.9 and 4.2 percent respectively. It is interesting to see that the increase in performance is higher for $XLM_{MLM+TLM}$ which might be motivated by the fact of stronger English performance of $XLM_{MLM+TLM}$ which serves as a teacher in XD. We can also compare XD based on $XLM_{MLM}$ with $XLM_{MLM+TLM}$ baseline as being to zero-shot cross-lingual transfer methods that work on top of $XLM_{MLM+TLM}$. It can be seen that $XLM_{MLM}$ + XD only slightly outperforms $XLM_{MLM+TLM}$. However, both methods combine effective. It is interesting to see that $XLM_{MLM+TLM}$ based XD also outperforms previous IndT state-of-the-art which uses labels for synthetically translated training set data. See Table 3 for full XD results.

---

[2]http url will be available after de-anonymization

[3]https://github.com/huggingface/pytorch-transformers

[4] https://github.com/allenai/allennlp

Table 3: Results of cross-lingual knowledge distillation (XD) for all languages simultaneously, compared to multilingual tuning (MLT) and individual tuning (IndT, results of tuning a separate model for each language by Lample & Conneau (2019)). Results for the two XLM varieties (MLM and TLM) are shown separately. Zero-shot scores (with no directly supervised tuning performed for these languages) are shown in gray.

| | en | fr | es | de | el | bg | ru | tr | ar | vi | th | zh | hi | sw | ur | avg |
|---|---|---|---|---|---|---|---|---|---|---|---|---|---|---|---|---|
| $XLM_{MLM}$ | | | | | | | | | | | | | | | | |
| Baseline | 83.7 | 76.0 | 76.8 | 73.8 | 73.2 | 73.8 | 72.1 | 66.9 | 68.6 | 71.9 | 67.6 | 71.5 | 64.3 | 64.4 | 61.3 | 70.7 |
| MLT | 83.3 | 79.4 | 80.0 | 78.5 | 78.8 | 79.2 | 76.8 | 74.2 | 75.0 | 75.8 | 75.0 | 78.0 | 71.8 | 71.5 | 66.6 | 76.6 |
| XD | 81.8 | 77.7 | 77.6 | 77.0 | 76.4 | 76.8 | 75.5 | 73.2 | 73.6 | 73.9 | 73.2 | 75.8 | 70.2 | 70.2 | 65.1 | 74.6 |
| $XLM_{MLM+TLM}$ | | | | | | | | | | | | | | | | |
| Baseline | 84.9 | 78.8 | 79.2 | 76.6 | 76.4 | 76.7 | 75.0 | 72.4 | 72.4 | 74.3 | 71.4 | 74.3 | 68.5 | 68.6 | 64.4 | 73.8 |
| MLT | 84.6 | 80.6 | 81.6 | 80.2 | 79.9 | 80.4 | 78.4 | 76.1 | 77.6 | 78.1 | 77.2 | 79.4 | 73.3 | 73.1 | 68.5 | 78.0 |
| IndT | 85.0 | 80.2 | 80.8 | 80.3 | 78.1 | 79.3 | 78.1 | 74.7 | 76.5 | 76.6 | 75.5 | 78.6 | 72.3 | 70.9 | 63.2 | 76.7 |
| XD | 83.9 | 80.7 | 80.4 | 79.3 | 79.3 | 79.4 | 77.4 | 75.4 | 77.4 | 77.2 | 76.1 | 78.9 | 72.5 | 71.6 | 67.7 | 77.0 |

Table 4: Results of cross-lingual knowledge distillation (XD) for specific languages ($l_2$) in combination with English, in comparison with true-label tuning (MLT) on the same language pair ($l_2$ and English). Results are shown only for English (source of labels), the language used for XD and the overall average for the sake of brevity.

| | en | fr | avg | en | sw | avg | en | ur | avg |
|---|---|---|---|---|---|---|---|---|---|
| $\mathbf{XLM}_{MLM}$ | | $l_2=$ French | | | $l_2=$ Swahili | | | $l_2=$ Urdu | |
| Baseline | 83.7 | 76.0 | 70.7 | 83.7 | 64.4 | 70.7 | 83.7 | 61.3 | 70.7 |
| MLT, en+$l_2$ | 82.0 | 78.7 | 71.5 | 82.7 | 71.1 | 72.4 | 79.5 | 65.0 | 70.8 |
| XD, en+$l_2$ | 82.9 | 78.7 | 72.2 | 82.7 | 71.4 | 72.5 | 82.5 | 64.0 | 71.1 |
| $\mathbf{XLM}_{MLM+TLM}$ | | $l_2=$ French | | | $l_2=$ Swahili | | | $l_2=$ Urdu | |
| Baseline | 84.9 | 78.8 | 73.8 | 84.9 | 68.6 | 73.8 | 84.9 | 64.4 | 73.8 |
| SepT | 85.0 | 80.2 | 76.7 | 85.0 | 70.9 | 76.7 | 85.0 | 63.2 | 76.7 |
| MLT, en+$l_2$ | 84.7 | 81.0 | 76.0 | 84.8 | 72.8 | 75.3 | 83.3 | 67.3 | 74.2 |
| XD, en+$l_2$ | 85.2 | 81.4 | 75.5 | 85.0 | 72.9 | 75.8 | 84.5 | 66.9 | 75.1 |

Finally, we observe that XD is behind our MLT training scheme where all languages are used. In the next subsection we provide comparisons between both our methods for distinct languages.

### 3.3.2 CASE STUDIES

In this set of experiments we first tune XLM on English and target language at the same time (MLT) and then compare it with XD where we distill knowledge from target language to English while not using labels from target language.

In spite of our expectations that MLT will perform better (based on results from previous subsection) the results show better performance of XD in most cases. XD shows better average performance on case of all 3 languages for $XLM_{MLM}$ and in case of Swahili and Urdu for $XLM_{MLM+TLM}$. When comparing by the target language solely MLT outperforms XD only for Urdu.

This suggests that XD is competitive with multilingual tuning when compared in more restricted cases. One possible explanation is that when tuning on true labels from all languages data, model has more freedom to ignore information coming from low-quality inputs like Urdu ones. When we train the model to imitate teacher logits we are more restrictive in terms of how the model can represent the data including low-quality examples. Refer to the Table 4 for comparison.

### 3.3.3 COMBINING XD WITH MULTILINGUAL TUNING

Finally, we take our insights from previous experimental results and try to come up with combined model that outperformes XD and multilingual tuning individually. Concretely, we take our bast MLT model (the one that was trained without Urdu data) and perform XD procedure on top of it. In this case both student and teacher initialization become stronger so we expect further increase after distillation performance.

Table 5: Results of cascaded combination of our best insights: multilingual tuning on all languages but Urdu with cross-lingual knowledge distillation (XD) on specific language sets; bests result gives multilingual tuning with XD for English, German, French, and Spanish on top

| | en | fr | es | de | el | bg | ru | tr | ar | vi | th | zh | hi | sw | ur | avg |
|---|---|---|---|---|---|---|---|---|---|---|---|---|---|---|---|---|
| MLT$_{no-ur}$ | 85.2 | 81.3 | 82.5 | 80.5 | 80.8 | 81.2 | 79.3 | 76.7 | 77.8 | 78.5 | 77.5 | 80.2 | 73.9 | 73.0 | 70.1 | 78.9 |
| MLT$_{no-ur}$ + XD fr | 84.0 | 81.3 | 81.5 | 79.9 | 79.7 | 81.0 | 78.4 | 75.6 | 77.3 | 77.8 | 76.8 | 79.5 | 72.9 | 72.2 | 69.6 | 78.0 |
| MLT$_{no-ur}$ + XD sw | 84.7 | 80.8 | 82.2 | 80.0 | 79.8 | 80.8 | 78.3 | 76.0 | 77.8 | 78.2 | 77.5 | 79.5 | 73.2 | 72.6 | 70.2 | 78.4 |
| MLT$_{no-ur}$ + XD ur | 82.4 | 78.8 | 79.6 | 78.3 | 78.0 | 78.6 | 76.6 | 72.8 | 75.1 | 75.7 | 74.5 | 77.6 | 70.0 | 70.8 | 67.4 | 76.5 |
| MLT$_{no-ur}$ + XD w/0 ur | 84.2 | 80.4 | 81.0 | 79.6 | 79.0 | 78.9 | 77.3 | 74.1 | 76.6 | 77.7 | 76.3 | 78.6 | 72.5 | 71.2 | 68.8 | 77.3 |
| MLT$_{no-ur}$ + XD 4-lang | 85.3 | 81.7 | 82.7 | 81.4 | 80.4 | 81.1 | 79.5 | 76.4 | 78.5 | 78.7 | 78.1 | 80.1 | 73.6 | 73.9 | 69.7 | 79.2 |

Table 6: Comparison between related work and our methods: multilingual tuning (MLT) and cross-lingual knowledge distillation (XD).
[†] denotes concurrent work, [♯] denotes a zero-shot approach that uses English training sentence translations, but not their labels.

| | en | fr | es | de | el | bg | ru | tr | ar | vi | th | zh | hi | sw | ur | avg |
|---|---|---|---|---|---|---|---|---|---|---|---|---|---|---|---|---|
| **Translate-train, tuned separately or multilingually** | | | | | | | | | | | | | | | | |
| Lample & Conneau (2019) | 85.0 | 80.2 | 80.8 | 80.3 | 78.1 | 79.3 | 78.1 | 74.7 | 76.5 | 76.6 | 75.5 | 78.6 | 72.3 | 70.9 | 63.2 | 76.7 |
| Park et al. (2019) | – | 78.7 | 78.2 | 76.4 | 76.7 | 75.8 | 75.5 | 73.3 | 73.7 | 74.2 | 72.3 | 74.3 | 72.2 | 71.6 | **71.3** | – |
| Wu & Dredze (2019) | 82.1 | 76.9 | 78.5 | 74.8 | 72.1 | 75.4 | 74.3 | 70.6 | 70.8 | 67.8 | 63.2 | 76.2 | 65.3 | 65.3 | 60.6 | 71.6 |
| Huang et al. (2019)[†] | **85.6** | 81.1 | 82.3 | 80.9 | 79.5 | **81.4** | **79.7** | 76.8 | 78.2 | 77.9 | 77.1 | **80.5** | 73.4 | 73.8 | 69.6 | 78.5 |
| MLT | 84.6 | 80.6 | 81.6 | 80.2 | 79.9 | 80.4 | 78.4 | 76.1 | 77.6 | 78.1 | 77.2 | 79.4 | 73.3 | 73.1 | 68.5 | 78.0 |
| MLT$_{no-ur}$ | 85.2 | 81.3 | 82.5 | 80.5 | **80.8** | 81.2 | 79.3 | 76.7 | 77.8 | 78.5 | 77.5 | 80.2 | **73.9** | 73.0 | 70.1 | 78.9 |
| MLT$_{no-ur}$ + XD$_{4-lang}$ | 85.3 | **81.7** | **82.7** | **81.4** | 80.4 | 81.1 | 79.5 | 76.4 | **78.5** | **78.7** | **78.1** | 80.1 | 73.6 | **73.9** | 69.7 | **79.2** |
| **Zero-shot: no labels from target language(s) used** | | | | | | | | | | | | | | | | |
| Conneau et al. (2018) | 73.7 | 70.4 | 70.7 | 68.7 | 69.1 | 70.4 | 67.8 | 66.3 | 66.8 | 66.5 | 64.4 | 68.3 | 64.2 | 61.8 | 59.3 | 67.2 |
| Schuster et al. (2019) | 73.7 | 65.1 | 67.4 | 64.4 | – | – | – | – | – | – | – | – | – | – | – | – |
| Chidambaram et al. (2018) | 71.6 | 64.4 | 65.2 | 65.0 | – | – | – | – | – | – | 62.8 | – | – | – | | – |
| Artetxe & Schwenk (2018) | 73.9 | 71.9 | 72.9 | 72.6 | 73.1 | 74.2 | 71.5 | 69.7 | 71.4 | 72.0 | 69.2 | 71.4 | 65.5 | 62.2 | 61.0 | 70.2 |
| Wu & Dredze (2019) | 82.1 | 73.8 | 74.3 | 71.1 | 66.4 | 68.9 | 69.0 | 61.6 | 64.9 | 69.5 | 55.8 | 69.3 | 60.0 | 50.4 | 58.0 | 66.3 |
| Lample & Conneau (2019) | 85.0 | 78.7 | 78.9 | 77.8 | 76.6 | 77.4 | 75.3 | 72.5 | 73.1 | 76.1 | 73.2 | 76.5 | 69.6 | 68.4 | 67.3 | 75.1 |
| Huang et al. (2019)[†] | **85.1** | 79.0 | 79.4 | 77.8 | 77.2 | 77.2 | 76.3 | 72.8 | 73.5 | 76.4 | 73.6 | 76.2 | 69.4 | 69.7 | 66.7 | 75.4 |
| MLT$_{no-ur}$ | – | – | – | – | – | – | – | – | – | – | – | – | – | – | **70.1** | – |
| XD$_{all}$ [♯] | 83.9 | **80.7** | **80.4** | **79.3** | **79.3** | **79.4** | **77.4** | **75.4** | **77.4** | **77.2** | **76.1** | **78.9** | **72.5** | **71.6** | 67.7 | **77.0** |

First we perform XD on French / English / Urdu separately. In all cases average performance drops which suggests that cross-lingual transformation of XLM that happens as a part of XD process is not effective in average. It is interesting to see that it also does not help even in case of target languages. We then try to remove Urdu from XD procedure since we observed that removing it from MLT significantly improves results but again get score below MLT baseline. We suppose that MLT, being already strong cross-lingual model, learned to handle low-quality inputs from low-resources and unrelated languages in specific ways. By introducing XD from these low quality sources, the regularities inside the model also break. That is why in our final experiment with used only data of high-quality from related languages such as German, French, and Spanish which gave as performance 0.3 points higher then MLT w/0 ur baseline (refer to Table 5 for comparison).

## 4 RELATED WORK

Many related research efforts use parallel data in addressing the task of cross-lingual sentence representation alignment. One such line of work aligns individual language models to the fine-tuned English LM (Conneau et al., 2018; Aldarmaki & Diab, 2019a;b). These approaches also use continuous sentence vector representations (embeddings) but require a pretrained monolingual language model for each language individually. This is impractical for large-scale state-of-the art transformers and as a consequence authors operate on small and weak language models achieving sub-optimal performance. Our approach only requires a single polyglot model to be trained.

Another line of work attacks the problem from a different angle by aligning sentence representations inside a polyglot model at pretraining time. Chidambaram et al. (2018) train multitask systems

where they consider isolated pairs of languages and jointly learn the tasks of source and target language modelling as well as sentence representation alignment.

Multilingual transformer language models (like mBert[5] (Devlin et al., 2018) or $XLM_{MLM}$ (Lample & Conneau, 2019)) can be viewed as methods where word representations are implicitly aligned at pretraining time through sharing of input wordpieces. In $XLM_{MLM+TLM}$ the authors reinforce this alignment by including pairs of parallel sentences as an additional BERT training objective. Work by Ren et al. (2019) is another example of explicit cross-lingual pretraining. Artetxe & Schwenk (2018), similarly to McCann et al. (2017), use a fixed-vector encoder-decoder machine translation setup to obtain sentence representations. Finally, concurrently to our work, Huang et al. (2019) present Unicoder LM, where they add 3 additional pretraining-time cross-lingual objectives on top of XLM. Their model is 1.6% behind our zero-shot self-teaching results; moreover, Unicoder LM can be used as a new baseline for our method (instead of $XLM_{MLM+TLM}$) which can potentially lead to even better results. Refer to Table 5 for the performance of the above-mentioned methods on XNLI.

On the other hand, a number of authors experimented with the use of labeled synthetic XNLI data from multiple languages as part of a single polyglot LM (multilingual tuning). Mulcaire et al. (2019) trained a polyglot ELMO model for word tagging tasks, while Wu & Dredze (2019) trained a stronger mBert model on all XNLI synthetic data. Their results however fall behind single-language XLM baselines and the authors did not compare their system with the single-language mBert baselines. Park et al. (2019) use adversarial examples to improve multilingual tuning on synthetic data for pairs of languages. Lastly, concurrent work (Huang et al., 2019) discover that multilingual tuning of a polyglot LM not only reduces the need for 15 individual LMs but also improves accuracy on average. Authors further push this result by fine-tuning Unicoder on all synthetic XNLI data. Their best non-zero-shot result is still 0.4% behind ours where we remove weakly translated data from an unrelated language from the tuning scheme. Moreover, our finding is again fully complementary with Unicoder multi-language fine-tuning, and can be easily combined for the best performance.

Finally, there is related work that applies Knowledge Distillation (Hinton et al., 2015) for big LMs or NLI. For example Tsai et al. (2019) train a small mBERT model with a goal to reduce the size of a large-scale polyglot network trained by Devlin et al. (2018) while Liu et al. (2019) apply KD in the context of multi-task learning where English NLI is one of the tasks in hand.

## 5 CONCLUSIONS

We presented our work on multilingual sentence representations and their application in cross-lingual natural language inference. Our results show that a single model trained for all 15 languages in the XNLI dataset can achieve better results than 15 individually trained models, and get even better when unrelated poorly-translated languages are removed from the multilingual tuning scheme. Next, we introduced cross-lingual knowledge distillation (XD), where multilingual sentence representations inside the same model are aligned without the use of end-task labels. Using XD we outperformed the previous methods that also do not use target languages labels. A combination of both our approaches gives further improvements and reaches a new state-of-the-art on the XNLI dataset.

Our future work includes applying XD to other sentence-level tasks that can benefit from cross-lingual knowledge transfer. It would also be interesting to see if the overall performance drops if the knowledge distillation dataset is different from the tuning data.

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
