# OpenReview forum: "XD: Cross-lingual Knowledge Distillation for Polyglot Sentence Embeddings"
_ICLR.cc/2020/Conference — Reject_

### Official Review · AnonReviewer3 · 2019-10-17
**Official Blind Review #3**

**Rating:** 1

**Review:**

This paper proposes two improved strategies for fine-tuning XLM (a multilingual variant of BERT) for cross-lingual NLI. First of all, it shows that fine-tuning a single model on the combination of all languages (the original English data from MultiNLI and their MT translation into the rest of languages) performs better than fine-tuning a separate model for each language. Furthermore, they show that minimizing the L2 distance between the English training sentences and their MT translation into the rest of languages, which does not explicitly use any labels in the foreign languages and is presented as a way of performing cross-lingual knowledge distillation, also performs better than zero-shot transferring a regular model fine-tuned in English.

I think that the paper makes some interesting contributions and, in particular, I think that the finding that multilingual fine-tuning performs better than the standard approach of fine-tuning a separate model for each language is important. Nevertheless, I am not convinced that there is enough novelty and substance on this, I have some concerns on the evaluation, and I think that the overall presentation should also be improved:

- I am not convinced by the "knowledge distillation" approach. First, although I see the connection, I do not think that presenting this as "knowledge distillation" is consistent with the common use of this term in the literature. More importantly, I do not see what is the value of this approach considering that multilingual fine-tuning performs better, and combining them both does not bring any clear improvement. The authors motivate it as a form of performing zero-shot cross-lingual transfer as, unlike the multilingual fine-tuning, this approach does not use any label in the foreign languages. However, I am not convinced at all by this reasoning, as it still relies on the translation of the English labeled data into the other languages. So, from a practical perspective, it requires the exact same resources as the other approach, as you would always be able to use the English labels for the rest of the languages, while being more complex and worse.

- It looks like the IndT results, which is the real baseline, are taken from the original XLM paper, while the rest of the results come from the authors' own runs, who use a different implementation. I think that you should also report IndT results from your own runs to make sure that your improvements come from the actual method, and not from small implementation details.

- You are trying small variations of your method (e.g. removing a particular language from the multilingual training) to support your claims, and it is not clear if the (rather small) differences in the results are significant. It would be good if you at least run the baseline multiple times and show the variance.

- It is unfair to try so many variants of your method in the test set, and then pick the best and claim SOTA as done in Table 6. Your final system looks rather ad-hoc and arbitrary: it is doing multilingual fine-tuning in all languages but Urdu, and cross-lingual knowledge distillation in a subset of 4 languages out of 15. It might get SOTA results in this particular scenario, but what if we move to a different set of languages, a different task, or even a different test set?

- The authors claim that "Urdu (ur) is an unrelated language" and "Swahili (sw) is loosely between French and Urdu in terms of relatedness to English", which they use to justify why Urdu behaves differently in their experiments. I do not speak neither Swahili nor Urdu, but from what I know this statement looks wrong. Swahili and English belong to completely different language families, and from what I know their grammar is very different. In contrast, Urdu at least belongs to the Indoeuropean language family.

- This is not relevant at all, but I would suggest the authors to find a different acronym instead of XD, which happens to be a widely used emoticon. I assume that the authors deliberately made this choice thinking that it would be funny, but I just find it confusing to see XD all over the place in a formal paper.

**Experience Assessment:**

I have published in this field for several years.

**Review Assessment: Checking Correctness Of Derivations And Theory:**

N/A

**Review Assessment: Checking Correctness Of Experiments:**

I carefully checked the experiments.

**Review Assessment: Thoroughness In Paper Reading:**

I read the paper thoroughly.

---

> ### Author Response · Authors · 2019-11-15
> **We will work on the presentation!, comments on novelty, motivation and other details below:**
>
> This was very insightful, thank you very much for your review! Sorry for the hasty submission, we will work on the presentation and improve the writing and clarity of the paper.
>
> You are right about Urdu and Swahili, thank you for correcting us. Instead the higher translation quality (BLEU score) in XNLI for Swahili and lower score for Urdu are the more correct justification for the choices concerning these two languages.
>
> In our view the novelty here is that we present 2 approaches that achieve state-of-the-art results on XNLI (at the moment of the ICRL submission deadline): one using labels for all the languages, another one using only English labels. The combination between the multilingual approach and XD gives a further small improvement, probably because they are based on the same XNLI inputs, but we still added this as an additional experiment.
>
> The main motivation for our knowledge distillation approach is that it can be potentially applied to any unannotated data, as long as the data has some translations available (from a parallel corpus OR an from an MT engine). Although we did not test it on a different dataset, in that scenario it does require any labels on the new data -- also not for English, just the English labels on the original dataset (used for creating the multilingual based model).
>
> We will add the individual-tuning results done on our own runs, as you suggested. We will also add statistical significance results via bootstrapping and re-run the key models of the paper to estimate the variance, as you proposed.
>
> Concerning the choice of languages, it was mostly guided, not ad-hoc: for example, the removal of Urdu was driven by its lowest BLEU score of the MT system used to create the XNLI dataset; the 4 languages of the combination were indeed chosen without a thorough comparison (why not 3? 5?), but again based on their BLEU score and NLI performance of the baseline model. The idea was just to illustrate the sensitivity of XD to the quality of multilingual “alignment” data.
>
> We agree that a more detailed investigation of language combinations would be exciting, we did our best to explain the motivation for the choices we did in this paper and we hope to provide a more thorough comparison in a followup paper.

---

### Official Review · AnonReviewer2 · 2019-10-23
**Official Blind Review #2**

**Rating:** 6

**Review:**

First, the authors propose to train a model for natural language inference (NLI) on multiple languages simultaneously. In particular, they translate English examples into all target languages and fine-tune a pretrained language model on all thereby obtained data at once. This is different from the previous state-of-the-art approach which consisted of, after translating from English into target languages, fine-tuning one NLI model for each language individually. The authors show that their approach is superior to training individual models for each language. For evaluation, XNLI is used.

Second, they introduce cross-lingual knowledge distillation (XD), where the same polyglot model is used both as teacher and student across languages to improve its sentence representations without using the target task labels. The main idea is that the same sentence in all languages should receive output representations as similar as possible.

The paper seems okay to me and the experiments seem solid. However, the results are not particularly surprising and the methods are not very innovative. The writing could be improved.

This paper could further be improved in the following ways:
- A more detailed investigation which combination of languages improve performance (and why?).
- Similarly: A combination of MTL and XD doesn't seem straightforward. Why? What is learned?

Smaller comments:
- Articles are missing frequently (e.g., "we substitute the word prediction head with classification layer" -> "we substitute the word prediction head with a classification layer")
- Table 5: "w/0" -> "w/o"?
- Have you run any significance tests?

**Experience Assessment:**

I have published one or two papers in this area.

**Review Assessment: Checking Correctness Of Derivations And Theory:**

N/A

**Review Assessment: Checking Correctness Of Experiments:**

I assessed the sensibility of the experiments.

**Review Assessment: Thoroughness In Paper Reading:**

I read the paper at least twice and used my best judgement in assessing the paper.

---

> ### Author Response · Authors · 2019-11-15
> **Will improve the writing; some comments on significance, language choices and method combination:**
>
> Thank you for your review! We will make sure to improve the writing and clarity of the paper, sorry for making you read a hurried submission. We will definitely incorporate your smaller comments.
>
> We did not run any significance test for the submitted version; since it was suggested by several reviewers, we will do it via bootstrapping, as well as re-run the key models to estimate the variance of their results.
>
> We agree that a more detailed investigation of language combinations would be exciting, we did our best to explain the motivation for the choices we did in this paper. We hope to provide a more thorough comparison in the next paper.
>
> As for “Similarly: A combination of MTL and XD doesn't seem straightforward. Why? What is learned?”:
> In the case of MTL, data from low-resource unrelated (to English) languages were used just as well as data for resource-rich high-translation-quality languages. Our case studies with particular languages suggest that the quality of the parallel signal matters for XD. By using XD with only a high-quality parallel signal from high-resource languages we are able to further improve the system learned with multilanguage finetuning that used data from all sources.

---

### Official Review · AnonReviewer1 · 2019-11-03
**Official Blind Review #1**

**Rating:** 6

**Review:**

What is the task?
Multilingual natural language inference (NLI)

What has been done before?
Current state-of-the-art results in multilingual natural language inference (NLI) are based on tuning XLM (a pre-trained polyglot language model) separately for each language involved, resulting in multiple models.

What are the main contributions of the paper?
[Not novel] Significantly higher average XNLI accuracy with a single model for all 15 languages.
[Moderately novel] Cross-lingual knowledge distillation approach that uses one and the same XLM model to serve both as teacher (for English sentences) and student (for their translations into other languages). The approach does not require end-task labels and can be applied in an unsupervised setting

What are the main results?
 A single model trained for all 15 languages in the XNLI dataset can achieve better results than 15 individually trained models, and get even better when unrelated poorly-translated languages are removed from the multilingual tuning scheme.
 Using XD they outperformed the previous methods that also do not use target languages labels.

Weaknesses :
1. Combining XD with multilingual tuning is not effective in improving average results or even in case of target languages
2. Final system is adhoc as experiments on a particular set of languages have been used to support claims. For example, Urdu was excluded to get the best MLT model. Only 4 languages were used while combining XD and MLT
3. Findings, methods and experiments are not strongly novel.
4. Paper was not an easy read.

Strengths:
 Using XD they outperformed the previous methods that also do not use target languages labels.



**Experience Assessment:**

I have published one or two papers in this area.

**Review Assessment: Checking Correctness Of Derivations And Theory:**

N/A

**Review Assessment: Checking Correctness Of Experiments:**

I assessed the sensibility of the experiments.

**Review Assessment: Thoroughness In Paper Reading:**

I read the paper at least twice and used my best judgement in assessing the paper.

---

> ### Author Response · Authors · 2019-11-15
> **Will improve the writing; comments on novelty and language choices below:**
>
> Thank you for your review and sorry for making you read a rushed submission -- we will definitely improve the writing and clarity of the paper.
>
> In our view the novelty here is that we present 2 approaches that achieve state-of-the-art results on XNLI (at the moment of the ICRL submission deadline): one using labels for all the languages, another one using only English labels. The combination between the multilingual approach and XD gives a further small improvement, probably because they are based on the same XNLI inputs, but we still added this as an additional experiment.
>
> Concerning the choice of languages, it was mostly guided, not ad-hoc: for example, the removal of Urdu was driven by its lowest BLEU score of the MT system used to create the XNLI dataset; the 4 languages of the combination were indeed chosen without a thorough comparison (why not 3? 5?), but again based on their BLEU score and NLI performance of the baseline model. The idea was just to illustrate the sensitivity of XD to the quality of multilingual “alignment” data.

---

### Official Review · AnonReviewer4 · 2019-11-04
**Official Blind Review #4**

**Rating:** 3

**Review:**

The paper addressed multilingual natural language inference. The motivations of the authors are two folds: 1/ to have only one model for all the languages instead of one model per language and 2/ achieving good results in a zero shot setup where only the english labels are available.

Previous work proposed first to learn multilingual language model in a self-supervised manner for the initialization. Then, fine-tuning it on the final task, separately for each different language. It results in multiple models, one for each language.

The authors proposed to fine-tune the model with all the data from the different languages simultaneously, resulting in only one model with comparable results.

In addition, they also proposed a method based on distillation where only the english targets are used. While the model doesn't require the label data for the other languages, the scores remained similar to the previous experiments.

Finally the authors proposed to combine both methods. It compares favorably, obtaining 4 points of improvement over SOTA in the zeroshot setup.

Pros:
-the motivation for having only one model is interesting
-the results are promising

Cons:
-one of main motivation of the paper is to achieve zeroshot as opposed to previous work. To that purpose, the authors chose to keep only the translated non-en input and not the non-en target. In XNLI, all the non-en training data, including target, are automatically translated from the English data. Therefore, the authors did not use less human annotated data and their approche still requires automatic translation. Hence, the motivation to perform the task in a zero-shot scheme, as opposed to previous work, doesn't seem correct.
-the paper was not always easy to follow and would benefit from more clarity.
'large margin of 5.9 and 4.2 points' in 3.2.1, please add the reference to table 6.
-the method 'significantly' improved results. I didn't see any significance measurements and it would be important to add them.

Overall, using all the data together seems like a natural and effective approach to and achieves good results through only one model.
However, the motivation behind 'distillation' is to perform in a zeroshot scheme. It seems abusive to me since it actually requires the exact same amount of human labels than the previous works.

**Experience Assessment:**

I have published one or two papers in this area.

**Review Assessment: Checking Correctness Of Derivations And Theory:**

I carefully checked the derivations and theory.

**Review Assessment: Checking Correctness Of Experiments:**

I assessed the sensibility of the experiments.

**Review Assessment: Thoroughness In Paper Reading:**

I read the paper thoroughly.

---

> ### Author Response · Authors · 2019-11-15
> **Will add significance and improve the writing; some comments on zero-shot inside:**
>
> Thank you for your review!
>
> We will definitely improve the writing and clarity of the paper, sorry for making you read a rushed submission. Thank you for the specific improvement suggestions, we will surely incorporate them.
>
> You are correct that our method has its differences in comparison to other zero-shot approaches in terms of the parallel signal we are using. However, our alignment approach (XD) does not use human labels in the XNLI dataset -- neither English, nor other languages, only a model trained on the annotated English data. Thus, the method can also be potentially applied to any unannotated data, as long as the data has some translations available (from a parallel corpus OR an from an MT engine). So it still belongs to the family of methods that don't use labels for target languages. We already highlighted the differences with other zero-shot methods in the results table and in the text and we will make sure to make it even more clear in the paper.
>
> As for the significance of the comparisons, we initially used the term generally, not in the statistical sense, when referring to the larger gap comparisons (like individually tuned vs our final approach combination). However, we will add statistical significance tests via bootstrapping, and we will also do a couple of random restarts for the key models of the paper to estimate the variance of their results.

---

### Author Response · Authors · 2019-09-27
**Code and model sharing**

We could not guarantee anonymity in the shared code (license, Google Drive user visibility) so we had to switch the sharing off till de-anonymization.

---

### Decision · Program_Chairs · 2019-12-19

**Decision:**

Reject

**Comment:**

This paper proposes a method for transferring an NLP model trained on one language a new language, without using labeled data in the new language.

Reviewers were split on their recommendations, but the reviews collectively raised a number of concerns which, together, make me uncomfortable accepting the paper. Reviewers were not convinced by the value of the experimental setting described in the paper—at least in the experiments conducted here, the claim that the model is distinctively effective depend on ruling out a large class of models arbitrarily. it would likely be valuable to find a concrete task/dataset/language combination that more closely aligns with the motivations for this work, and to evaluate whether the proposed method is genuinely the most effective practical option in that setting. Further, the reviewers raise a number of points involving baseline implementations, language families, and other issues, that collectively make me doubt that the paper is fully sound in its current form.